# The risk of bleeding and perforation from sigmoidoscopy or colonoscopy in colorectal cancer screening: A systematic review and meta-analyses

Isabella Skaarup Kindt[1]◐*, Frederik Handberg Juul Martiny[1,2]◐, Emma Grundtvig Gram[1,3], Anne Katrine Lykke Bie[1‡], Christian Patrick Jauernik[1‡], Or Joseph Rahbek[1‡], Sigrid Brisson Nielsen[1‡], Volkert Siersma[1], Christine Winther Bang[1], John Brandt Brodersen[1,3,4]

1 The Centre of General Practice, Department of Public Health, University of Copenhagen, Copenhagen, Denmark, 2 Department of Social Medicine, Bispebjerg and Frederiksberg Hospital, Copenhagen, Denmark, 3 The Research Unit for General Practice in Region Zealand, Region Zealand, Denmark, 4 The Research Unit for General Practice, Department of Community Medicine, Faculty of Health Sciences, The Arctic University of Norway, Tromsø, Norway

◐ These authors contributed equally to this work.
‡ AKLB, CPJ, OJR and SBN also contributed equally to this work.
* krc330@sund.ku.dk

**Data Availability Statement:** All relevant data is available on OSF (https://osf.io/jyh98/).

## Abstract

### Introduction

Physical harm from Colorectal Cancer Screening tends to be inadequately measured and reported in clinical trials. Also, studies of ongoing Colorectal Cancer Screening programs have found more frequent and severe physical harm from screening procedures, e.g., bleeding and perforation, than reported in previous trials. Therefore, the **objectives** of the study were to systematically review the evidence on the risk of bleeding and perforation in Colorectal Cancer Screening.

### Design

Systematic review with descriptive statistics and random-effects meta-analyses.

### Methods

We systematically searched five databases for studies investigating physical harms related to Colorectal Cancer Screening. We assessed the internal and the external validity using the ROBINS-I tool and the GRADE approach. Harm estimates was calculated using mixed Poisson regression models in random-effect meta-analyses.

### Results

We included 89 studies. Reporting and measurement of harms was inadequate in most studies. In effect, the risk of bias was critical in 97.3% and serious in 98.3% of studies. All GRADE ratings were very low. Based on severe findings with not-critical risk of bias and 30

**Funding:** One of the first authors (FHJM) received financial support via the research grant "Sara Krabbes legat" from the Danish Society for General Practitioners (https://www.dsam.dk/forskning/sara_krabbes_legat/), covering expenses related to Open Access publication. The Danish Cancer Society Research Center (https://www.cancer.dk/forskning/stoette-til-forskning/funding/) funded one year's salary for FHJM to conduct the systematic review, and the William Demant Foundation (https://www.williamdemantfonden.dk/) supported FHJM's participation in the Preventing Overdiagnosis Conference 2017 in Quebec, Canada. The funders had no role in study design, data collection and analysis, decision to publish, or preparation of the manuscript. The first author is independent of the funding bodies.

**Competing interests:** We have no conflicts of interest to disclose. This does not alter our adherence to PLOS ONE policies on sharing data and materials.

days follow-up, the risk of bleedings per 100,000 people screened were 8 [2;24] for sigmoidoscopy, 229 [129;408] for colonoscopy following fecal immunochemical test, 68 [39;118] for once-only colonoscopy, and 698 [443;1045] for colonoscopy following any screening tests. The risk of perforations was 88 [56;138] for colonoscopy following fecal immunochemical test and 53 [25;112] for once-only colonoscopy. There were no findings within the subcategory severe perforation with long-term follow-up for colonoscopy following any screening tests and sigmoidoscopy.

## Discussion

Harm estimates varied widely across studies, reporting and measurement of harms was mostly inadequate, and the risk of bias and GRADE ratings were very poor, collectively leading to underestimation of harm. In effect, we consider our estimates of perforation and bleeding as conservative, highlighting the need for better reporting and measurement in future studies.

## Trial registration

**PROSPERO registration number:** CRD42017058844.

## 1. Introduction

Colorectal Cancer Screening (CRCS) can, like any other screening program, cause unintended harm, including physical and psychosocial harm [1, 2]. Evidence has shown that the categorization of the unintended harms of CRCS lacks consensus [3]. However, there is an agreement that the most serious type of harm in CRCS is physical harms [2]. Several countries have implemented CRCS, where people receive either a sigmoidoscopy or a colonoscopy as a stand-alone intervention or following other screening tests, e.g. colonoscopy following fecal immunochemical test (FIT) [4, 5]. These CRCS programs aim to detect pre-cancer lesions or colorectal cancer at an localized stage to reduce mortality and morbidity [5–7].

Studies of ongoing CRCS programs have found that severe physical complications to sigmoidoscopy and colonoscopy, e.g., bleeding and perforation, are more frequent and severe than previous clinical trials have suggested [8–11]. In addition, clinical trials have had a tendency to present the harm of CRCS in an unbalanced manner compared to the benefits of screening, and sometimes completely omit or disregard the reporting of harm [4, 11–13]. Inadequate reporting of harms of CRCS is potentially compounded when systematic reviews do not pay sufficient attention to the issues concerning measurement and reporting of harms in clinical trials. This concern led to the publication of the PRISMA-harms extension to support more rigor in systematic reviews of adverse events of medical interventions [14]. However, former systematic reviews of CRCS, even those published after the PRISMA-harms extension, have not referenced it [15–23]. In effect, the harms of CRCS may be underreported in clinical trials and in former systematic reviews compared to the real-world rate of harms in ongoing CRCS programs. In addition, the methodological quality of the evidence about harm of CRCS have received little attention and consequently the trustworthiness of the evidence is uncertain.

Therefore, we conducted a systematic review according to recommendations within the PRISMA-harms extension, aiming to assess the quality of the evidence in the area and the real-

world risk of all types of physical harms related to CRCS [24]. We found surprisingly a hetero-geneous evidence base concerning the assessment, definition, measurement, and reporting of physical harms related to CRCS. Therefore, we had to divide the review into separate studies to allow adequate attention to the findings (S1 Appendix in S1 File) [24]. Here, we report find-ings from studies that assessed two of the most severe procedure-related physical harms of CRCS, i.e., bleeding and perforation. Our aims were fourfold. First, we aimed to investigate how studies measured and reported bleeding and perforation. Second, to assess the internal and the external validity of findings in studies. Third, to quantify the risk and the consequences of bleeding and perforation related to CRCS and fourth, to describe characteristics of the screening intervention and setting of the screening population that might modify the risk of bleedings and perforations or the consequences thereof.

## 2. Methods

Here we outline the key methodological aspects of the review with a detailed account available in the protocol, which was registered before the conduct of the systematic review at PROS-PERO: CRD42017058844 [25].

### 2.1 Study eligibility

Two reviewers independently assessed the eligibility of each of the identified studies, extracted data from included studies, subcategorized bleeding and perforation events, and assessed the internal and external validity of these findings from studies. Discrepancies were discussed in pairs of two until consensus, potentially involving a third review author in case of disagree-ments. Studies were eligible if they investigated the risk of bleeding or perforation during CRCS using sigmoidoscopy or colonoscopy for the general population, i.e., asymptomatic adults (+18 years of age) at average risk of colorectal cancer. We accepted minor deviations from the inclusion criteria's (S2 Appendix in S1 File).

### 2.2 Search strategy & information sources

We searched six databases: PubMed, MEDLINE, Embase, CINAHL, PsycINFO and the Cochrane Library on the 12th of April 2017 with an updated search on the 4th of March 2022. We used backtracking in included studies to identify studies potentially missed by the search strategy (S3 Appendix in S1 File).

### 2.3 Study selection

Studies were included, regardless of study design, risk of bias, year of publication and language [26]. Study authors were contacted if full text studies were not available or in case of doubt about inclusion. We provided reasons for all studies excluded at full text level (S4 Appendix in S1 File)

### 2.4 Data extraction process

The data extraction template was inspired by the PRISMA-harms extension [16] and a generic data collection template from the Cochrane Collaboration [26]. Data extraction included information about the study, i.e., study characteristics, details about the screening intervention and any information about physical harms (S5 Appendix in S1 File).

## 2.5 The internal validity–The ROBINS-I tool

We used the ROBINS-I tool to assess the internal validity of findings, i.e., bias assessment of outcome level [27]. The ROBINS-I tool includes seven bias domains: confounding bias, inception bias, misclassification bias, performance bias, missing data bias, measurement bias, and reporting bias [27, 28]. We did not assess the domain confounding bias, because no information of a control group was available. We assessed the risk of bias: low, moderate, serious and critical, and the likely direction of the effect bias might have on the outcome: unpredictable, underestimation, and overestimation [27].

## 2.6 The external validity–The GRADE approach

We assessed the external validity of findings without critical risk of bias, using the GRADE approach [27, 29]. One reviewer graded the evidence with subsequent validation by a second review author. The external validity (GRADE rating) of the evidence was graded: high, moderate, low, or very low. Studies that were one-armed started at "low quality" and was further downgraded either -1 or -2 based on assessment of four domains: 1) the risk of bias, 2) inconsistency of results, 3) imprecise results, and 4) publication bias [29]. We did not assess the fifth GRADE domain, indirectness of the evidence, due to very strict eligibility criteria, so indirect evidence was not included for review. The evidence was upgraded +1 or +2 according to three criteria: 1) large magnitude of effect, 2) adequate precision of the effect size or 3) reason to believe the outcome was caused by screening and no other factors (low risk of confounding) (S6 Appendix in S1 File). Further, we noted the overall score of the GRADE rating in evidence profile tables, e.g., if the evidence concerning mild bleedings was rated down -2 due to risk of bias and -1 due to publication bias with upgrading +1 due to large magnitude of effect, the GRADE sum would be -3.

## 2.7 Categorization

**2.7.1 Categorization of procedures (Subpopulations).** We stratified the risk of bleeding and perforation on screening procedure: 1) sigmoidoscopy, 2) once-only colonoscopy, 3) colonoscopy following FIT, and 4) follow-up colonoscopy after sigmoidoscopy or other types of screening tests than FIT. We categorized screening procedures to promote homogeneity in analyses. When studies examined more than one screening procedure, e.g., one part of the population receiving sigmoidoscopy and the other part receiving once-only colonoscopy, we handled this as two separate subpopulations.

**2.7.2 Categorization of bleeding and perforation (Subcategories).** We subcategorized bleeding and perforation events according to the studies' definitions of severity and follow-up time, with inspiration from the ASGE lexicon (Subcategories) [30] (S7, S8 Appendices in S1 File). According to the definition and other information about harms reported in studies, we were able to categorize bleeding and perforation into three levels of severity:

1. Severe: Bleedings and perforations that required hospitalization, surgery, transfusion, or in other ways described as severe.

2. Mild: All bleedings and perforations that did not require hospitalization or prevent completion of the procedure, self-limiting events, harms described as mild, self-limiting, or the like.

3. Not Defined (ND): When severity of harms was not defined or if the definition was unclear.

Further we used the follow-up time reported in studies to categorize bleeding and perforation into three further levels:

1. Short term: Any bleeding or perforation that occurred immediately, during, or within 7 days.

2. Long term: Bleedings or perforations that occurred 0–30 days follow-up after the screening procedure. Studies with 30-day follow-up include follow-up from the time the procedure is performed to 30 days after the procedure.

3. Not Reported (NR): If follow-up time was not reported.

This leads to nine potential combinations of severity and follow-up subcategories for both bleeding and perforation (Subcategories).

## 2.8 Statistical method

We used Microsoft Excel for descriptive statistics [31] and the *R* software [32] to perform meta-analyses. A meta-analysis estimate of the risk was calculated in a Poisson regression model with a random-effect for subcategories to account for heterogeneity, and with the logarithm of subcategories size as offset. We performed meta-analyses stratified on screening procedures, follow-up time, severity, and the risk of bias (dichotomized: critical or not-critical). Furthermore, we did post-hoc meta-analyses, combining the three severity categories for perforation and bleeding stratified on screening procedures, and follow-up time. These post hoc meta-analyses were conducted to account for the interrelationship between mild and severe types of harms, e.g., screening procedures that cause many severe bleedings is likely to cause fewer mild bleedings. We used the Clopper-Pearson method to determine 95% confidence intervals (95% CI). The heterogeneity was quantified with $X^2$ and the $I^2$ [33]. We considered $I^2$ larger than 75% as considerable heterogeneity [33]. Consequences of harms were descriptively analyzed.

## 2.9 Synthesis of results

We quantified harms when possible and presented additional findings descriptively when meta-analyses were not justified, e.g., when the subcategory of the outcome of interest was only assessed for one subpopulation, in S9, S10 Appendices in S1 File. We present numbers as the risk of harm per 100,000 people screened. In studies that did not report the number of people screened, we imputed the number, using conversion factors calculated from studies that both reported the number of people screened and procedures performed (S11 Appendix in S1 File).

# 3. Results

## 3.1 Study selection

We identified 17,058 studies in the first search strategy and further 6,223 studies in the updated search. We included 134 studies for review of which 89 studies reported on bleeding or perforation (66.0%). Of these, 104 studies were identified through the search strategy and the remaining 30 studies were identified via backtracking. We excluded 262 studies after full-text reading (S2 File and Fig 1).

## 3.2 Study characteristics

We included 89 studies that reported on bleeding or perforation in this study. When accounting for more than one screening procedure in some studies, i.e., subpopulations, bleeding was

## PRISMA FLOW CHART

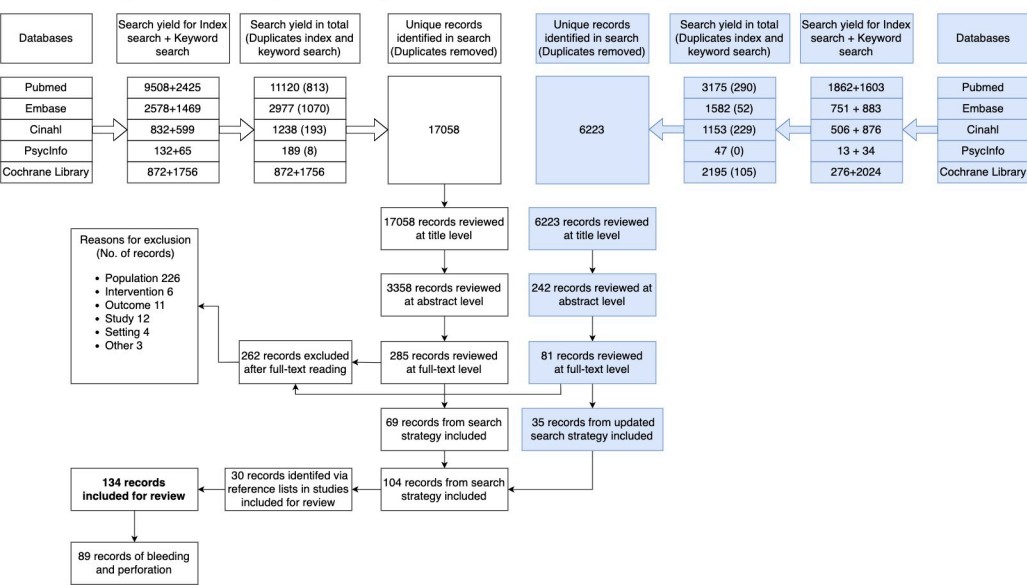

**Fig 1. PRISMA flowchart.** Study selection process.

assessed in 104 subpopulations (69.0%) and perforation was assessed in 105 subpopulations (70.0%). There were 123 combinations of subcategories of bleeding and 108 combinations of subcategories of perforation when accounting for multiple assessments with varying follow-up time and severities of the outcome for some subpopulations (S12 Appendix in S1 File).

**3.2.1 Characteristics of RCTs and NRSs.** Included studies were both RCTs and NRSs and less than half of the studies reported on sociodemographic information (Table 1).

**3.2.2 Characteristics of procedure groups.** Across subpopulations, the most widely used procedure was colonoscopy following FIT; bleeding 44 (29.0%) and perforation 45 (30.0%). Sigmoidoscopy was the least used procedure; bleeding 13 (9.0%) and perforation 11 (7.0%). The provision of polypectomies was more common in groups where people received once-only colonoscopy; bleeding 22 (69.0%) and perforation 23 (68.0%) and colonoscopy following FIT; bleeding 32 (73.0%) and perforation 29 (64.0%), than sigmoidoscopy and colonoscopy following any screening tests. 82.0% of subpopulations reported that polypectomies were performed but none of the subpopulations using sigmoidoscopy as procedure reported the rate of polypectomies (S13 Appendix in S1 File).

**Table 1. Key characteristics of included studies that assessed bleeding or perforation.**

| Study characteristics | Studies N = 89 (%) |
|---|---|
| **Study design (RCT)** | 21 (18.7%) |
| **Sociodemographic information reported** | 19 (21.0%) |
| **Studies with minor deviations from eligibility criteria** | 21 (34.0%) |
| **Age deviation** | 11 (18.0%)[a] |
| **Increased risk of CRC** | 12 (20.0%)[a] |

[a]Some studies deviated from our eligibility criteria in regard to age and increased risk of CRC.

**Table 2. Overview of worst-bias scores in total for subcategories with assessment of perforation or bleeding.**

| | Low | Moderate | Serious | Critical | Total |
|---|---|---|---|---|---|
| Perforation | 0 (0.0%) | 3 (2.7%) | 50 (46.3%) | 55 (51.0%) | 108 (100%) |
| Bleeding | 0 (0.0%) | 2 (1.6%) | 59 (47.9%) | 62 (50.4%) | 123 (100%) |

## 3.3 Measurement and reporting of bleeding and perforation

We identified 64 distinct definitions of bleeding and 36 of perforation across the 104 and 105 subpopulations, respectively (S7, S8 Appendices in S1 File). We did not perform meta-analyses of the subcategory short-term events, as very few (6.0%) subpopulations had short-term follow-up for both bleeding and perforation. Instead, we clustered the harm subcategories concerning follow-up time: short-term events and NR events together in the category NR to avoid losing information from subpopulations with follow-up time that was either short-term or NR (S12 Appendix in S1 File). To sum up, we subcategorized bleeding and perforation into six and five subcategories respectively. Harms were subcategorized as ND for 24.0% of subpopulations with assessment of bleeding and 44.0% of perforation. Less than half of the included studies reported on details about measurements including follow-up time, outcome assessor, and measurement tool (S14, S15 Appendices in S1 File).

## 3.4 The internal validity

None of the subpopulations had low risk of bias. The majority of the assessments of bleeding (51.0%) and perforation (50.4%) had critical risk of bias. This was mainly due to the risk for missing data bias and measurement bias, e.g., due to lack of dropout analyses and inadequate attempts to measure harms (Table 2) (S16, S17 Appendices in S1 File).

## 3.5 The external validity

We evaluated studies that did not have critical risk of bias, and all had "very low" quality, corresponding to a score below zero (S18, S19 Appendices in S1 File). In the GRADE ratings, we reached a "floor effect" concerning the sum score of the up- and downgrading factors in the GRADE ratings of the evidence (Table 3).

**Table 3. GRADE ratings of the evidence for perforation and bleeding.**

| Perforation | | | | |
|---|---|---|---|---|
| | Sigmoidoscopy | Colonoscopy following FIT | Once-only colonoscopy | Colonoscopy following any screening tests |
| Severe-NR | NA | NA | -3 | NA |
| Severe-longterm | NA | -4 | -3 | NA |
| ND-longterm | -4 | -1 | -3 | -3 |
| Mild-longterm | -4 | -1 | -2 | -2 |
| ND-NR | -4 | -2 | -2 | -3 |
| **Bleeding** | | | | |
| | Sigmoidoscopy | Colonoscopy following FIT | Once-only colonoscopy | Colonoscopy following any screening tests |
| Severe-NR | -2 | -2 | -2 | -2 |
| Severe-longterm | -3 | -6 | -6 | -3 |
| ND-longterm | -4 | -2 | -5 | -5 |
| Mild-NR | NA | -1 | -2 | -4 |
| Mild-longterm | -2 | -4 | -5 | NA |
| ND-NR | -4 | -4 | -5 | NA |

NA: Not Applicable due to lack of studies

**Table 4. Point estimates per 100,000 screened people for bleeding.**

| Procedure | Stratification | Total bleedings—longterm | Total bleedings—NR | Severe-longterm | Severe-NR | Mild-longterm | Mild-NR | ND-longterm | ND-NR |
|---|---|---|---|---|---|---|---|---|---|
| Colonoscopy following any screening tests | Not-critical risk of bias | 675 [448;1015] | 780 [447;1264] | 698 [443;1045] | 439 [201;831] | – | 341 [137;702] | 0 [0;3240] | – |
| | Critical risk of bias | 186 [44;649] | 333 [147;754] | 47 [15;147] | 108 [67;173] | 1205 [329;3056] | 676 [192;2381] | 128 [21;787] | 324 [231;456] |
| | All studies | 205 [65;644] | 372 [178;774] | 198 [36;1082] | 170 [87;330] | 1205 [329;3056] | 562 [208;1515] | 113 [18;708] | 324 [231;456] |
| Sigmoidoscopy | Not-critical risk of bias | 25 [5;134] | 31 [19;52] | 8 [2;24] | 30 [15;52] | y189 [149;237] | – | 2 [0;7] | 43 [9;126] |
| | Critical risk of bias | 799 [413;1391] | 35 [1;834] | – | 0 [0;13] | 8 [4;16] | 104 [63;173] | 799 [413;1391] | 0 [0;2225] |
| | All studies | 56 [9;362] | 46 [6;336] | – | 1 [0;32421] | 40 [4;356] | 104 [63;173] | 29 [1;1530] | 15 [0;3474] |
| Once-only colonoscopy | Not-critical risk of bias | 202 [106;386] | 145 [106;197] | 68 [39;118] | 167 [152;183] | 320 [97;1059] | 24 [9;52] | 420 [248;711] | 150 [87;257] |
| | Critical risk of bias | 648 [69;6119] | 130 [84;202] | 42 [14;99] | 23 [2;226] | – | 87 [84;91] | 1314 [128;13544] | 155 [119;201] |
| | All studies | 268 [106;676] | 140 [111;177] | 63 [39;103] | 72 [24;219] | 258 [97;685] | 62 [28;137] | 461 [85;2504] | 154 [113;211] |
| Colonoscopy following FIT | Not-critical risk of bias | 436 [304;624] | 1156 [202;6629] | 229 [129;408] | 757 [327;1486] | 631 [440;905] | 7600 [7371;7833] | 793 [687;914] | 210 [5;1165] |
| | Critical risk of bias | 674 [593;764] | 340 [338;342] | 674 [593;764] | 281 [112;705] | – | 80 [26;186] | – | 443 [160;1232] |
| | All studies | 445 [315;627] | 399 [203;782] | 247 [142;429] | 324 [145;728] | 631 [440;905] | 776 [31;19343] | 793 [687;914] | 416 [160;1087] |

Not applicable due to no studies for analysis in the respective subcategory

The worst possible GRADE rating -6 for bleeding and -4 for perforation. The best grading was -1 corresponding to "very low" quality. We downgraded all analyses -2 due to serious risk of bias in more than half of the studies. We rarely downgraded due to inconsistency of results because of small differences in effect estimates, or imprecision because most of the analyses had adequate sample size. We downgraded all analyses -1 due to publication bias.

## 3.6 The risk of bleeding and perforation

Meta-analyses are presented in Tables 4 and 5, with forest plots available in S1–S3 Figs.

**3.6.1 Meta-analyses for bleeding.** Across the four screening procedure groups, the event rate per 100,000 people screened for bleeding ranged from 31–1156 within seven days to 25–675 within 30 days. Judging the studies that did not have critical risk of bias, the risk of bleeding was highest for colonoscopy following any screening tests 675 [448;1015], while the risk was lowest for sigmoidoscopy 25 [5;134] within 30 days. Across all screening procedures and subcategories of bleeding, the risk ranged from 0–7600. In most analyses we found a trend towards lower harm estimates in analyses of findings with critical risk of bias compared to studies not-critical risk of bias, e.g., 186 [44;649] bleedings per 100,000 people screened compared to 675 [448;1015] for colonoscopy following any screening tests within 30 days. However, in other analyses we found the reverse trend, e.g., the NR+shortterm had more events than long term in the total assessment for colonoscopy following any test and following FIT.

**3.6.2 Meta-analyses for perforation.** Across the four screening procedure groups, the event rate per 100,000 people screened for perforation ranged from 4–117 within seven days

**Table 5. Point estimates per 100,000 screened people for perforation.**

| Procedure | Stratification | Total perforations–longterm | Total perforations–NR | Severe-longterm | Severe-NR | Mild-longterm | ND-longterm | ND-NR |
|---|---|---|---|---|---|---|---|---|
| Colonoscopy following any screening tests | Not-critical risk of bias | 117 [44;313] | 32 [1;1270] | – | – | 121 [33;310] | 0 [0;3240] | 32 [1;1270] |
| | Critical risk of bias | 46 [15;140] | 115 [56;233] | 42 [13;138] | 39 [16;94] | – | 43 [3;552] | 147 [145;150] |
| | All studies | 59 [26;134] | 100 [50;201] | 42 [13;138] | 39 [16;94] | 121 [33;310] | 40 [3;595] | 121 [58;256] |
| Sigmoidoscopy | Not-critical risk of bias | 4 [1;10] | 2 [0;14] | – | – | 8 [2;24] | 2 [0;7] | 2 [0;14] |
| | Critical risk of bias | 10 [0;56] | 5 [0;84] | – | – | 10 [0;56] | – | 5 [0;84] |
| | All studies | 4 [2;611] | 5 [1;36] | – | – | 9 [3;23] | 2 [0;7] | 5 [1;36] |
| Once-only colonoscopy | Not-critical risk of bias | 50 [49;50] | 31 [9;109] | 53 [25;112] | 70 [5;937] | 430 [317;579] | 22 [0;1532] | 12 [2;43] |
| | Critical risk of bias | 21 [10;46] | 20 [13;32] | – | 24 [15;39] | – | 21 [10;46] | 8 [2;43] |
| | All studies | 39 [38;39] | 22 [10;46] | 53 [26;97] | 23 [16;34] | 430 [317;579] | 25 [9;72] | 12 [10;15] |
| Colonoscopy following FIT | Not-critical risk of bias | 80 [59;107] | 53 [26;105] | 88 [56;138] | – | 59 [45;78] | 70 [63;79] | 53 [25;105] |
| | Critical risk of bias | 292 [239;353] | 87 [72;107] | 292 [239;353] | 128 [67;246] | – | – | 85 [69;104] |
| | All studies | 85 [62;115] | 83 [69;101] | 97 [62;152] | 128 [67;246] | 59 [45;78] | 70 [63;79] | 81 [66;98] |

– Not applicable due to no studies for analysis in the respective subcategory

and 2–53 within 30 days. Judging the studies that did not have critical risk of bias, the risk of perforation was highest for colonoscopy following FIT 53 [26;105], while the risk was lowest for sigmoidoscopy 2 [0;14] within 30 days. Across all screening procedures and subcategories of perforation, the risk ranged from 0–430. In most analyses we found a trend towards lower harm estimates in analyses of findings with critical risk of bias compared to studies with not-critical risk of bias, e.g., 46 [15;140] perforations per 100,000 people screened compared to 117 [44;313] for colonoscopy following any screening tests within 30 days. However, in other analyses we found the reverse trend, e.g., the NR+shortterm had more events than long term in the total assessment for colonoscopy following any test and following Sigmoidoscopy.

## 3.7 The consequences of bleeding and perforation

The consequences of bleeding were reported for 39 (36.0%) subpopulations. We could categorize consequences of bleeding into three groups: 1) need of transfusion, 2) other treatment, and 3) hospitalization. Transfusion was the most frequently reported consequence and was reported in 23 subcategories (18.7%) (S20 Appendix in S1 File). Of note, information about the prognosis of patients, e.g., sequelae of treatments, number of hospital days, complications arising during transfusion etc., were seldom reported.

The consequences of perforation were reported for 33 (40.0%) subpopulations. We could categorize consequences of perforation into four groups: 1) death, 2) treatment, 3) morbidity, and 4) requiring hospitalization. In four subcategories (3.7%) perforation resulted in death. In 22 (20.4%) subcategories, participants underwent treatment, and in two (1.8%) cases perforation caused morbidity (S21 Appendix in S1 File). In 75 (60.0%) subcategories, there were no available information on the consequences of perforation.

### 3.8 Factors potentially modifying the risk of harm or the consequences of perforation or bleeding

In total, potential modifiers were reported for 69 (30.0%) subpopulations. The most frequently investigated modifiers were polypectomy rate, age, sex, and expertise of the endoscopists. Polypectomy was investigated as a modifier in 23 (28.0%) subcategories for bleeding and 24 (26.0%) subcategories for perforation. Polypectomies had a statistically significant effect on the occurrence of the outcome in 19 (4.0%) subcategories for bleeding and 21 (5.0%) subcategories for perforation. The risk of bleeding and perforation increased with age, polypectomy and inversely with expertise of the endoscopists (S22, S23 Appendices in S1 File).

## 4. Discussion

### 4.1 Summary of main findings

We included 89 studies for review. Measurement and reporting of bleeding and perforation were heterogeneous across studies, and less than half of the included studies reported details about measurements including follow-up time, outcome assessor, and measurement tools used. The internal validity of findings from studies was very low with critical risk of bias in more than half of studies both concerning estimates of bleeding and perforation. We did not find a clear dosis-response pattern between the risk of harm and the risk of bias and we did not find any systematic differences between the harm estimates from RCTs versus observational studies. Further, the external validity was very low for all analyses with further downgrading in most analyses. We found that participation in CRCS programs entails an increased risk of bleeding and perforation events, especially in older people and if polypectomy was performed. Based on severe findings with not-critical risk of bias and 30 days of follow-up, the risk of bleedings per 100,000 people screened were 8 [2;24] for sigmoidoscopy, 229 [129;408] for colonoscopy following FIT, 68 [39;118] for once-only colonoscopy, and 698 [443;1045] for colonoscopy following any screening tests. Similarly, the risk of perforations was 88 [56;138] for colonoscopy following FIT and 53 [25;112] for once-only colonoscopy. There were no findings within the subcategory severe perforation with long-term follow-up for colonoscopy following any screening tests and sigmoidoscopy. Few studies assessed factors potentially modifying the risk of harm or the consequences thereof. The consequences of harms were seldom reported with information about consequences for bleeding in 36.0% of studies and perforation in 40.0% of studies. Further, information about the consequences of harms was sparse.

### 4.2 Strengths and limitations

Our findings are based on a rigorous systematic review process, which followed the best available guidance for systematic reviews of adverse events of medical interventions from the Cochrane Collaboration's Handbook [26], the PRISMA 2020 guideline [34], the PRISMA-harms extension [14] and the AMSTAR checklist [35].

This review did not account for physical harms that occurred as a result of treatment of screen-detected lesions, except the immediate removal of polyps or adenomas during the screening procedure, i.e., polypectomy, or surveillance resulting from screening. Therefore, the true risk of bleeding and perforation of all steps of the screening cascade in CRCS programs is likely higher than reported here [2, 36]. Of note, our findings should be interpreted with care because studies lacked a control group and had high risk of bias.

Studies rarely reported definitions, follow-up time, and severity of harms and generally these were reported in very non-specific terms. Therefore, our subcategorizations are most

likely subject to misclassification. This might also explain the large heterogeneity in meta-analyses and the reason why we do not find any consistent trend in association between risk estimate and risk of bias. Further, 39 subcategories for bleeding and perforation reported zero events with doubtful attempts to measure harms and a narrow definition of harm. Such studies might bias the overall harm estimate towards the null. Conversely, it might be argued that studies that did assess harmful events, which did not occur, e.g., zero bleedings, may not report this finding. However, due to the poor measurement and reporting in general, and the general tendency for studies not to report zero findings, we consider it more likely that the overall estimate of harms is biased towards the null. Of note, the reverse might hold true, and this judgment is based on our reading of the literature in the area.

With our subcategorization of harms, we get a more detailed overview of the severity of harm and follow-up time compared to other studies. We believe that it is easier to separate and interpret these risk estimates, but whether we subcategorize the harms in the proper categories is debatable. A possible explanation for the fact that we do not see a higher risk after 30 days across screening procedures could be due to the ND-long-term category, which potentially decreases the risk, as the severity of harm is undefined or narrow. In addition, as mentioned, the dichotomization of the risk of bias may also contribute to that we do not see a higher risk of harm after 30 days among studies with critical risk of bias. Our categorization of harm can be challenging to compare with other studies' narrow categorizations of harm, which is why we may risk seeing fewer mild events and many severe events and vice versa across the screening procedures, which can give a distorted picture of which screening methods that should be recommended as the primary screening procedure.

Any bias assessment may overlook, underestimate, or overestimate the risk of bias. In the dichotomization of the risk of bias into critical vs. not-critical, we did not account for the fact that most studies considered as not-critical were of serious risk of bias. Therefore, the studies with not-critical risk of bias were also generally of low quality. Therefore, the comparison between not-critical and critical studies might not reflect the true effect that bias might have on effect estimates, i.e., comparisons between moderate and critical risk of bias studies might have shown other trends than our bias comparisons. However, we judged that there were too few studies with moderate risk of bias to make a post hoc analysis. We found that the GRADE approach was difficult to apply to the heterogeneous evidence on physical harms of screening. Therefore, we predetermined thresholds in the assessment of GRADE, to improve the applicability of the GRADE approach in this setting. This was at the cost of reducing the comparability of our GRADE ratings to other ratings.

## 4.3 Findings compared to former systematic reviews

We found six former systematic reviews that assessed both bleeding and perforation [15–17, 19, 21, 22] (S24 Appendix in S1 File. Estimates of bleedings and perforations varied both between former reviews and compared to the present review. These differences are likely due to the large heterogeneity in studies that were included for review and how outcomes were defined.

Four former reviews have categorized the severity of bleedings as severe if the event required hospitalization or medical intervention similar to our categorization. Only one review included mild bleedings [17], and three reviews excluded events considered as mild [14, 15, 21]. Excluding mild bleedings can contribute to the underestimation and underreporting of bleeding as a harm of CRCS. Another former review, which did not define the severity of bleeding, found that the risk was 5 [2;9] bleedings per 100,000 screened people with once-only

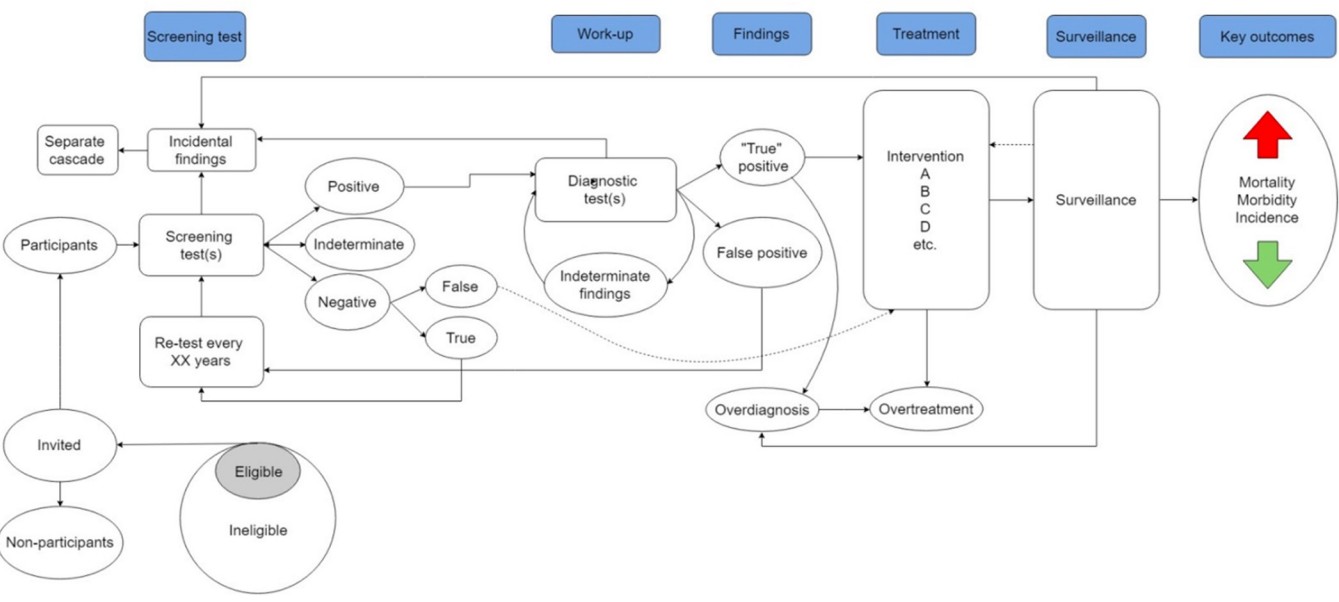

**Fig 2. The screening cascade.**

colonoscopy compared to 268 [106;676] bleedings in the present review [21] (S25 Appendix in S1 File).

Only one former review categorized the severity of perforation, where perforation was categorized as a severe complication. Here, the risk of severe perforations was 59 [37;89] per 100,000 screened people with colonoscopy following FIT [19] compared to 97 [62;152] perforations per 100,000 people screened in the present review. The remaining five reviews did not categorize the severity of perforation. Another former review, found that the risk was 61 [10;111] perforations per 100,000 screened people with colonoscopy following FIT compared to 85 [62;115] perforations in the present review [17] (S26 Appendix in S1 File).

All reviews claim to implicitly assess the harms of screening for the entire screening cascade. However, we argue that none of the former reviews, and neither ours yet explicitly, included an assessment that covered all steps in the screening cascade [36] (S2 File and Fig 2).

### 4.4 Implications for future research

Although there are benefits from CRCS, our findings highlight that more and better studies are needed about the adverse effects of screening programs to ensure a balanced evidence base [16]. However, until we have a thorough and good evidence base for the harms of CRCS, we consider it challenging to discuss whether the benefits outweigh the harms of CRCS and whether implementation actually improves public health. The heterogeneous definitions and inadequate methodological approaches to measure and report bleeding and perforation of CRCS leads to results, that do not truly reflect the actual frequency or severity of these harms. Future studies on CRCS would benefit from adhering to guidelines that clearly define and conceptualize the potential harms of CRCS and provide criteria for measurement of harms, e.g., the ASGE-lexicon [9, 30]. Of note, only one NRS study (0.7%) used a guideline on how to categorize the severity of bleeding and perforation [9, 30]. None of the NRSs referred to the most commonly used STROBE guideline for reporting of harms in NRSs, and no extension of the guideline is currently available [37]. In addition, none of the included RCTs referred to the CONSORT-harms extension [38]. Former systematic reviews seem to compound inadequate

reporting of physical harms due to a lack of focus on measurement and reporting of harms in original studies [14]. Therefore, our findings indicate that there is a need for authors of future systematic reviews to follow PRISMA-harms [34]. In line with this, trialists conducting RCTs about CRCS could use the CONSORT-harms extension [38] and it would likely improve harms measurement and reporting in NRSs if the STROBE guideline had a similar extension [37]. We used the ROBINS-I tool, which is currently the best available tool to assess the internal validity in studies. In addition, we used the GRADE approach, which is widely recommended. However, we found that both tools had to be amended quite extensively for the purposes of our review, i.e., in a setting of screening and adverse events. Yet, the approaches could only provide a rough distinction between good-quality and poor-quality studies. For future research, there is a need for better-developed tools in the field of harmful effects of screening [39]. This review provides a starting point for creating a more appropriate tool to assess the internal and external validity in studies. In addition, dissemination of our findings to clinicians and lay people would enable the incorporation of harmful effects into screening information materials, which could contribute to a more balanced communication about the benefits and harms of screening [40, 41].

## 5. Conclusion

We found various and unclear definitions of bleedings and perforations in terms of assessments methods, follow-up time, and severity of harm. Further, studies had low internal and external quality and high heterogeneity when pooled in meta-analyses. Based on severe findings with not-critical risk of bias and 30 days follow-up time, we found that the risk of bleedings and perforations varied significantly between the four screening procedures compared to former systematic reviews in the area. Our risk estimates varied widely across subcategories as well in the post hoc analyses. This might be due to our subcategorization of harm and the dichotomization of the risk of bias. Therefore, our risk estimates are likely to be conservative and underestimated due to studies inadequate attempts to measure and report harms of CRCS [8, 10, 11, 42]. Due to the variation in the analyses of the risk of bleeding and perforation between subcategories of harms and critical versus not critical studies we cannot conclude with certainty that one of the four screening procedures are more or less safe.

In comparison with former systematic reviews, we found higher risk estimates for bleeding and perforation. However, former reviews excluded mild bleedings and perforations, contributing to the fact that our subcategorization of harms could be challenging to compare with former reviews harm assessments. Given the above, there is a need for better evidence that take measurement and reporting of bleeding and perforation during CRCS, and in general screening programs, into account. In addition, we need to modify existing tools, i.e., ROBINS-I and the GRADE approach, or develop tools specifically for studies that assess harms to make them applicable for the heterogeneous and often low-quality evidence about harms.

## Supporting information

**S1 Checklist. PRISMA-HARMS checklist (completed).**
(DOCX)

**S1 File.** Appendix 1 –Deviations from the published protocol in the systematic review, Appendix 2 –Study eligibility, Appendix 3 –Search strategy & information sources, Appendix 4 –Reasons for all studies excluded (total list), Appendix 5 –Data extraction templates, Appendix 6 –The GRADE approach, Appendix 7 –Subcategories of bleeding, Appendix 8 –Subcategories of perforation, Appendix 9 –Study characteristics of special case studies and studies with an

unscreened control group, Appendix 10 –Characteristics of additional subpopulations, Appendix 11 –Conversion factor for each procedure group, Appendix 12 –Combination of subcategories. Appendix 13 –Characteristics of procedure groups, Appendix 14 –Adequacy of harm measurement across studies for bleeding, Appendix 15 –Adequacy of harm measurement across studies for Perforation, Appendix 16 –Bias distributions across all studies that assess bleeding, Appendix 17 –Bias distributions across all studies that assess perforation, Appendix 18 –Characteristics of the external validity for bleeding, Appendix 19 –Characteristics of the external validity for perforation, Appendix 20 –The consequences of bleeding, Appendix 21 – The consequences of perforation, Appendix 22 –Factors potentially modifying occurrences of bleeding. Appendix 23 –Factors potentially modifying occurrences of perforation, Appendix 24 –Bleeding and perforation assessed in six former systematic reviews, Appendix 25 –Comparison between former systematic reviews that assess bleeding and current review, Appendix 26 – Comparison between former systematic reviews that assess perforation and current review. (PDF)

**S2 File. Overview over all studies included for review.**
(PDF)

**S3 File. General information about the systematic review.**
(DOCX)

**S4 File. PROSPERO protocol.**
(PDF)

**S1 Fig. All meta-analyses of all types of bleeding and perforation events.**
(PDF)

**S2 Fig. All meta-analyses of subcategories for bleeding.**
(PDF)

**S3 Fig. All meta-analyses of subcategories for perforation.**
(PDF)

## Acknowledgments

A special thanks to the people listed below for their help assessing the eligibility of studies published in other languages than Scandinavian and English. We are grateful to our colleagues listed below for their support with assessing studies' eligibility and translating relevant publications.

| Country | Name | Title(s) | Affiliations |
|---|---|---|---|
| Germany | David Klemperer | Prof. Dr. | Ostbayerische Technische Hochschule Regensburg Faculty of Social and Health Care Sciences Seybothstraße 2 D-93053 Regensburg Germany |
| Poland | Maciek Godycki-Cwirko | Prof. MD. PhD | Centre for Family and Community Medicine, Medical University of Lpdz, Poland |
| Brazil | Janos Valery | MD | PhD student at Department of Preventive Medicine FMUSP |
| Italy | Giuseppe Febbo | Dr. | Movimento Giotto |
| France | Jean Yes Le Reste | Pr. Dr. | EA 7479 SPURBO faculté de médecine et des sciences de la santé, université de bretagne occidentale, Brest, France |
| Slovenia | Mateja Bulc | MD, PhD, specialist in GP | Assoc. Prof. at Department of Family Medicine at Medical Faculty of Ljubljana University |
| The Czech Republic | Bohumil Seifert | Assoc. prof. Dr.Ph. D. | Charles University, First Faculty of Medicine, Department of General Practice, Prague, Czech republic |
| Japan | Ryuki Kassai | Prof. | Department of Community and Family Medicine, Fukushima Medical University, 1 Hikarigaoka, Fukushima, Fukushima, 960–1295 Japan. |

## Author Contributions

**Conceptualization:** Frederik Handberg Juul Martiny, John Brandt Brodersen.

**Data curation:** Isabella Skaarup Kindt, Frederik Handberg Juul Martiny, Emma Grundtvig Gram, Anne Katrine Lykke Bie, Christian Patrick Jauernik, Or Joseph Rahbek, Sigrid Brisson Nielsen.

**Formal analysis:** Isabella Skaarup Kindt, Frederik Handberg Juul Martiny, Emma Grundtvig Gram, Anne Katrine Lykke Bie, Christian Patrick Jauernik, Or Joseph Rahbek, Sigrid Brisson Nielsen, Volkert Siersma, Christine Winther Bang, John Brandt Brodersen.

**Funding acquisition:** Frederik Handberg Juul Martiny, John Brandt Brodersen.

**Investigation:** Isabella Skaarup Kindt, Frederik Handberg Juul Martiny, Emma Grundtvig Gram, Anne Katrine Lykke Bie, Christian Patrick Jauernik, Or Joseph Rahbek, Sigrid Brisson Nielsen, Volkert Siersma, Christine Winther Bang, John Brandt Brodersen.

**Methodology:** Isabella Skaarup Kindt, Frederik Handberg Juul Martiny, Emma Grundtvig Gram, Anne Katrine Lykke Bie, Christian Patrick Jauernik, Or Joseph Rahbek, Sigrid Brisson Nielsen, Volkert Siersma, Christine Winther Bang, John Brandt Brodersen.

**Project administration:** Frederik Handberg Juul Martiny, John Brandt Brodersen.

**Resources:** Frederik Handberg Juul Martiny, John Brandt Brodersen.

**Software:** Frederik Handberg Juul Martiny, Volkert Siersma, Christine Winther Bang.

**Supervision:** Volkert Siersma, John Brandt Brodersen.

**Validation:** Isabella Skaarup Kindt, Frederik Handberg Juul Martiny, Emma Grundtvig Gram, Anne Katrine Lykke Bie, Christian Patrick Jauernik, Or Joseph Rahbek, Sigrid Brisson Nielsen, Volkert Siersma, Christine Winther Bang, John Brandt Brodersen.

**Visualization:** Isabella Skaarup Kindt, Frederik Handberg Juul Martiny, Emma Grundtvig Gram, Volkert Siersma, Christine Winther Bang.

**Writing – original draft:** Isabella Skaarup Kindt.

**Writing – review & editing:** Isabella Skaarup Kindt, Frederik Handberg Juul Martiny, Emma Grundtvig Gram, Anne Katrine Lykke Bie, Christian Patrick Jauernik, Or Joseph Rahbek, Sigrid Brisson Nielsen, Volkert Siersma, Christine Winther Bang, John Brandt Brodersen.

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
