## [Decision Letter · Decision Letter 0]

25 May 2023

PONE-D-22-34695The risk of physical harms from sigmoidoscopy or colonoscopy in colorectal cancer screening: a systematic review with meta-analyses on the risk of bleeding and perforationPLOS ONE

Dear Dr. Kindt,

Thank you for submitting your manuscript to PLOS ONE. After careful consideration, we feel that it has merit but does not fully meet PLOS ONE’s publication criteria as it currently stands. Therefore, we invite you to submit a revised version of the manuscript that addresses the points raised during the review process.

We look forward to receiving your revised manuscript.

Kind regards,

Sanjiv Mahadeva, MRCP, MD

Academic Editor

PLOS ONE

Journal Requirements:

All authors have completed the ICMJE uniform disclosure form at www.icmje.org/coi_disclosure.pdf and declare: the first author had financial support from the Danish Society for General Practice, the Danish Cancer Society and the William Demant Foundation for the submitted work, there were no financial relationships with any organizations that might have an interest in the submitted work in the previous three years; no other relationships or activities that could appear to have influenced the submitted work.

Please respond by return email with your amended Competing Interests Statement and we will change the online submission form on your behalf.

3. We noted in your submission details that a portion of your manuscript may have been presented or published elsewhere. [DETAILS AS NEEDED] Please clarify whether this [conference proceeding or publication] was peer-reviewed and formally published. If this work was previously peer-reviewed and published, in the cover letter please provide the reason that this work does not constitute dual publication and should be included in the current manuscript.

Additional Editor Comments:

This is an important systematic review & the data is of interest to the GI community. However, some corrections are needed.

In line with the Reviewer's comments (see attached Word document), please consider changing the Title accordingly.

Reviewers' comments:

Reviewer's Responses to Questions

**Comments to the Author**

1. Is the manuscript technically sound, and do the data support the conclusions?

Reviewer #1: Yes

2. Has the statistical analysis been performed appropriately and rigorously? 

Reviewer #1: Yes

3. Have the authors made all data underlying the findings in their manuscript fully available?

Reviewer #1: Yes

4. Is the manuscript presented in an intelligible fashion and written in standard English?

Reviewer #1: Yes

5. Review Comments to the Author

Reviewer #1: Kindt and Martiny plus colleagues present an interesting systematic review on physical harms following endoscopy (plus or minus biopsies) of the colon in humans. In a very detailed and coherently reported systematic review they develop a new overview on how such harms can be separated in severity and proximity to the endoscopy as well as how often bleeding and perforation occurs. Moreover, they meta-analyse the literature and report that the occurrence of these complications seems to be more often occurring than described in previous attempts to systematically review comparable or similar topics. The messages are in general clearly presented and the topics described in detail. This systematic review provides important insights into the inadequate reporting and measurement of harms in studies of colorectal cancer screening and highlights the need for better reporting and measurement of harms in future studies. In the file I managed to download, I only have access to the first 8/25 appendixes.

Please see attachment.

6. PLOS authors have the option to publish the peer review history of their article (what does this mean?). If published, this will include your full peer review and any attached files.

Reviewer #1: **Yes: **Christian Gluud

---

## [Author Response · Author response to Decision Letter 0]

18 Aug 2023

Responses to the reviewer:

Thank you for providing us the opportunity to revise our manuscript entitled "The risk of physical harms from sigmoidoscopy or colonoscopy in colorectal cancer screening: a systematic review with meta-analyses on the risk of bleeding and perforation". We are grateful to the reviewer for the careful evaluation of our manuscript. In the revised version of the manuscript with tracked changes, we have attempted to take the concerns raised by the reviewer into account. Please find a point-to-point reply to the reviewer’s comments in the table below. 

Kindt and Martiny plus colleagues present an interesting systematic review on physical harms following endoscopy (plus or minus biopsies) of the colon in humans. In a very detailed and coherently reported systematic review they develop a new overview on how such harms can be separated in severity and proximity to the endoscopy as well as how often bleeding and perforation occurs. Moreover, they meta-analyse the literature and report that the occurrence of these complications seems to be more often occurring than described in previous attempts to systematically review comparable or similar topics. The messages are in general clearly presented and the topics described in detail. This systematic review provides important insights into the inadequate reporting and measurement of harms in studies of colorectal cancer screening and highlights the need for better reporting and measurement of harms in future studies. In the file I managed to download, I only have access to the first 8/25 appendixes.

Thank you for your kind comments to our work. We have double-checked the uploaded appendices and we have access to all 25 appendices and now 26.

Reviewer’s comments answered one by one:

Questions/Comments from the reviewer Answers from the authors

1. The Prospero protocol and the systematic review seem a bit like representing two universes. Have I missed an appendix entitled: changes from protocol to the review? I have full understanding that a protocol cannot be fully compatible with a systematic review after several hundred of studies and six or more years of work. I suggest that the authors develop an appendix that tries to describe the modifications that have been made on the journey so one understands the developments and potential weaknesses. Thank you for your suggestion, which we have followed. We have added an appendix that describes the deviations from the protocol during the review process to facilitate a better understanding of the review process and its potential weaknesses. The new appendix is placed in the supplementary material as "Appendix 1 – Deviations from the published protocol in the systematic review”. In the revised manuscript file, the appendix is mentioned in the last paragraph of the introductory section. 

2. In the protocol, the authors write: “In summary, we have prioritized to include comparative studies to allow for comparison with existing reviews and to facilitate comparison of harm and harm assessment to benefit.” I fully accept that the authors do not assess benefits, but in their discussion they need to have a paragraph dealing with the balance between harms and benefits. The endoscopies are conducted to identify tumors and later to try to treat such tumors. I know that it has been hard to prove benefits of endoscopies, but think that the more recent trials and studies support some benefits. 

 Thank you, we agree that CRCS have benefits. We have added the following sentence “Although there are benefits from CRCS, our findings highlight that more and better studies are needed about the adverse effects of screening programs to ensure a balanced evidence base (16). However, until we have a thorough and good evidence base for the harms of CRCS, we consider it challenging to discuss whether the benefits outweigh the harms of CRCS and whether implementation actually improves public health.” to our discussion in the section 4.4.

3. I had to read the text several times and I did not become fully enlightened. What were the reasons for subdividing the endoscopy into ‘after FIT’ etc.? It should become fully clear why these subdivisions were conceived and if it is reasonable to keep them after the present finding. 

Moreover, how does the authors interpret similarities and dissimilarities in harm outcomes between the different subdivisions?

 Thank you for asking for clarification. We have amended our description of the reasons for subdividing into 4 procedure groups in section 2.7.1.

In brief, we have made this division for two reasons:

1) The included studies themselves follow this division. 

2) In order to make the groups as comparable as possible with regard to, among other things, risk, we chose to divide the groups into these four procedure groups. We found that there was a different case mix in the procedure groups, which is why a different division could potentially bias the result.

Regarding the question about interpretation of similarities or dissimilarities in harm outcomes between the four screening procedure groups, we have added a concluding remark about the observed variation in the second last paragraph of the conclusion. Further, in the submitted article we have provided a summary of the harm profile for each of the 4 screening procedure groups, and we have noted that misclassification might occur in the limitations section. 

4. In a similar vein, what was the reason for using the subdivision into short term and long term? 

And why is short-term part of long term? Maybe the readability could be improved by deleting this rather not so informative latter division?

 We can understand why this might seem confusing. We will try to explain our rationale behind the choice.

We do not think it is fair to compare studies that have different follow-up time, because more harm will arguably occur during a 30-day follow-up period versus a 7-day follow-up period, because bleeding and perforations do not necessarily occur within the first 7 days.

In addition, we have used the follow-up time that is in studies and made what we thought were meaningful categories based on that. We have added an elaboration about long-term follow up to avoid confusion in section 2.7.2. 

5. For the outcome death, I understand that the included studies do in fact not report on deaths that often. I would assume that death (all-cause mortality) would have been a natural part of such a review, why not choose death as an outcome? When I read the protocol, the authors are talking about severe adverse events, and here death is a natural component. And for all composite outcomes, one should analyse also the individual components. Moreover, having death as a separate outcome, the authors will have better abilities to highlight the lousy reporting of this important outcome.

 We agree that death is an important outcome. Therefore, we chose to divide reporting of the review into the present publication, reporting findings related to bleeding and perforation, while deaths and cardiopulmonary events are reported in a separate publication. We have elaborated the rationale for this in the newly added appendix 1, which was requested by the reviewer. 

Therefore, we have not included death in this publication from the review. 

6. In the discussion, you do not mention the risks of psychological harms as a limitation when one decides only to look at physical harms.

 The purpose of the present review was explicitly to study physical harms of CRCS. Therefore, we do not find that the omission of quantifying psychosocial harms of CRCS is a limitation of our review. 

However, we agree that psychosocial harms related to CRCS are important. We have found that the quality of assessment tools to measure these are insufficient, which we have described in another systematic review, which was recently published (1).

We have described how many types of harm may occur during CRCS, and often that tools to measure these are lacking in the introduction.. 

7. In the discussion, the authors write: “Any bias assessment may overlook, underestimate, or overestimate the risk of bias.” This is correct. However, our primary interest should be ‘Any assessed bias risk may overestimate or underestimate the reported findings’ ─ and especially if higher risk of bias is associated with underestimation of harms?

 We agree that it is important to investigate the relationship between the risk of bias and the estimate of harm because one might think that studies with a higher risk of bias would find lower estimates of harm compared to studies with less bias (higher quality studies). We have described this relationship in the results section 3.6. In addition, we have added the following sentence in

summary of main finding: “We did not find a clear dose-response pattern between the risk of harm and the risk of bias”.

8. In an appendix, please compare observations from intervention groups of randomised trials to that of observational studies.

 Thanks for your suggestion. However, it was not part of the review aims, neither in the protocol, nor in the present publication, to compare the harm estimates in trials to harm estimates from observational studies. 

Bringing this to our attention. We will try to explain our rationale behind not making a new appendix where we compare observations from intervention groups of randomised trials to that of observational studies.

We regarded evidence from included RCTs as comparable to evidence from non-randomized studies since there was no data from the control group in RCTs, i.e., all studies are effectively 1-armed, which is why we have chosen to add them together and consider the RCTs as equal in terms of the quality of the evidence as in the observational studies. Instead, we have chosen to present our findings based on the 4 screening procedure groups further stratified according to 1) a dichotomized worst-score counts risk of bias assessment, i.e., critical vs. not critical risk of bias, and 2) outcome categories , which were based on severity and follow-up time.

9. Please do not use any abbreviations in the abstract. This has been corrected in the abstract.

10. The title “The risk of physical harms from sigmoidoscopy or colonoscopy in colorectal cancer screening: a systematic review with meta-analyses on the risk of bleeding and perforation” 

should rather become

“The risks of physical harms from sigmoidoscopy or colonoscopy in colorectal cancer screening: a systematic review of observational evidence”.

 Thank you for your suggestion to change the title. We can see where your suggestion comes from, as we do not use the control groups from the RCT studies and thus consider them observational studies. We will try to explain the rationale behind our article title below: 

1) We aim to estimate the risks for these two outcomes, not the evidence that these outcomes are more prevalent than in a control group. An estimate for risk does not become better if the cohort in which the risk is estimated is accompanied by a control group and/or the intervention is randomised.

2) PRISMA recommends that, if meta-analyses have been carried out, this should be stated in the title of the article. Therefore, we do not find it wise to delete "meta-analyses" from the title.

3) PRISMA also recommends that you write in the title which types of harms that have been studied. Therefore, we believe it is important to keep bleeding and perforation instead of the more general term “physical harms”. 

We suggest that we change the title to the following: “The risk of bleeding and perforation from sigmoidoscopy or colonoscopy in colorectal cancer screening: a systematic review and Meta-analyses.”

---

## [Decision Letter · Decision Letter 1]

29 Sep 2023

The risk of bleeding and perforation from sigmoidoscopy or colonoscopy in colorectal cancer screening: a systematic review and meta-analyses

PONE-D-22-34695R1

Dear Dr. Kindt,

We’re pleased to inform you that your manuscript has been judged scientifically suitable for publication and will be formally accepted for publication once it meets all outstanding technical requirements.

Kind regards,

Sanjiv Mahadeva, MRCP, MD

Academic Editor

PLOS ONE

Additional Editor Comments (optional):

Thank you for the revised manuscript, which is satisfactory. Please clarify the "FIT" abbreviation in the abstract for the final proof.

Reviewers' comments:

Reviewer's Responses to Questions

**Comments to the Author**

1. If the authors have adequately addressed your comments raised in a previous round of review and you feel that this manuscript is now acceptable for publication, you may indicate that here to bypass the “Comments to the Author” section, enter your conflict of interest statement in the “Confidential to Editor” section, and submit your "Accept" recommendation.

Reviewer #1: (No Response)

2. Is the manuscript technically sound, and do the data support the conclusions?

Reviewer #1: (No Response)

3. Has the statistical analysis been performed appropriately and rigorously? 

Reviewer #1: (No Response)

4. Have the authors made all data underlying the findings in their manuscript fully available?

Reviewer #1: (No Response)

5. Is the manuscript presented in an intelligible fashion and written in standard English?

Reviewer #1: (No Response)

6. Review Comments to the Author

Reviewer #1: (No Response)

7. PLOS authors have the option to publish the peer review history of their article (what does this mean?). If published, this will include your full peer review and any attached files.

Reviewer #1: No

---

## [Editor Report · Acceptance letter]

23 Oct 2023

PONE-D-22-34695R1 

The risk of bleeding and perforation from sigmoidoscopy or colonoscopy in colorectal cancer screening: a systematic review and meta-analyses 

Dear Dr. Kindt:

I'm pleased to inform you that your manuscript has been deemed suitable for publication in PLOS ONE. Congratulations! Your manuscript is now with our production department. 

Kind regards, 

on behalf of

Prof Sanjiv Mahadeva 

Academic Editor

PLOS ONE